# Slip Casting and Solid-State Reactive Sintering of BCZY(BaCe_x_Zr_0.9−x_Y_0.1_O_3−d_)-NiO/BCZY Half-Cells

**DOI:** 10.3390/membranes12020242

**Published:** 2022-02-19

**Authors:** Sandrine Ricote, Robert J. Kee, William G. Coors

**Affiliations:** 1Mechanical Engineering Department, Colorado School of Mines, 1500 Illinois Street, Golden, CO 80401, USA; rjkee@mines.edu; 2Hydrogen Helix, Golden, CO 80401, USA; grover.h2helix@icloud.com

**Keywords:** protonic ceramic, BCZY, solid-state reactive sintering, slip casting, brazing

## Abstract

Slip casting was used to prepare BaCe_x_Zr_0.9−x_Y_0.1_O_3−d_(BCZY)-NiO tubes with a diameter of ½ inches (1.25 cm) and ¾ inches (1.875 cm). Two compositions were studied: BCZY18 and BCZY27 for x = 0.1 and 0.2, respectively. The unfired tubes were then dip-coated with three layers of the BCZY electrolyte membrane. Solid-state reactive sintering was used, meaning that the support and membrane were prepared with the precursors (oxides and carbonates). After co-sintering at 1550 °C, a 20-micron thick dense BCZY layer was well-adhered to the 1 mm thick BCZY-NiO support, as confirmed by scanning electron microscopy. The sintered BCZY-NiO/BCZY tubes were sealed onto alumina or BCZY substrates using a silver-based braze (with TiO_2_ and CuO additions). Gas tightness was achieved under 2 bar when covering the silver braze with a ceramic (Resbond) sealing layer. These slip cast tubes are intended for use as hydrogen electrodes in various protonic ceramic devices, and the advantages of short tubes for reactor design are discussed.

## 1. Introduction

Significant manufacturing progress has been achieved since the discovery of high-temperature proton conductors in 1981 [1]. The most studied materials are from the solid solution of barium cerate and barium zirconate, either doped with yttrium (BaCe_1−x−y_Zr_x_Y_y_O_3−d_, BCZY) [2,3,4,5] or co-doped with yttrium and ytterbium (BaCe_1−x−y−z_Zr_x_Y_y_Yb_z_O_3−d_, BCZYYb) [6,7,8]. Sintering of doped barium zirconate requires a temperature above 1600 °C for tens of hours [9,10]. During such a heat treatment, barium evaporates from the perovskite structure, leading to a barium deficient composition, whose conductivity is significantly lower [11]. It is worth noting that barium cerates are easier to sinter. Sintering aids such as NiO and ZnO have proven effective in reducing the sintering temperature by 200–300 °C [11,12]. Another breakthrough was the discovery of solid-state reactive sintering [13,14,15]. This cost- and time-effective method combines the discrete steps of solid-state reaction and traditional powder sintering. A liquid phase forms above 1100 °C (from BaO resulting from the BaCO_3_ decomposition and NiO that is used as a sintering aid) and enhances the cation transport along grain boundaries.

High-temperature proton conductors, also referred to as protonic ceramics, can be used as electrolytes in protonic ceramic fuel cells (PCFCs) [6,7,8,16,17,18], protonic ceramic electrolysis cells (PCECs) [19,20,21], and membrane reactors [22,23,24,25,26], just to name a few applications. Much of the research on device testing has been performed on planar cells, but the tubular geometry can bring advantages such as easier sealing and greater robustness. Tubes can be fabricated by extrusion (open ends) or slip casting (closed ends). Table 1 compiles some of the research on tubular protonic-ceramic-based devices. 

One of the most common fabrication methods for the tubular support electrode is extrusion, as it leads to a uniform density distribution [33]. The BCZY powder (or the mixture of precursors for solid-state reactive sintering) and NiO are mixed with additives (dispersants, binders, plasticizers) and a solvent. The as-prepared paste is then extruded through a die (typically cylindrical) and dried [34]. The resulting tube is open on both ends. To prepare closed-end tubes, slip casting is a cost-effective technique. The first step consists of making a mold in gypsum. The water-based ceramic slip is then poured into the mold, and the water is sucked out by capillarity. After drying, the ceramic body can be removed from the mold and sintered [35].

In this contribution, closed-end tubes were slip cast with a mixture of precursors to form 35/65 wt.% BCZY-NiO after sintering. Two compositions were investigated: BCZY18 (BaCe_0.1_Zr_0.8_Y_0.1_O_3−d_) and BCZY27 (BaCe_0.2_Zr_0.7_Y_0.1_O_3−d_). A BCZY layer was dip-coated on the unsintered tubes and then co-sintered at 1550 °C. Silver-based brazing was developed to achieve a hermetic seal between the tubes and the support (either alumina or BCZY).

## 2. Materials and Methods

The molds for slip casting were prepared using Plaster of Paris and a ½-inch (1.25 cm) stainless steel arbor, as illustrated in Figure 1. Larger tubes were also prepared with a larger arbor: ¾-inch (1.875 cm) diameter.

BCZY18(BaCe_0.1_Zr_0.8_Y_0.1_O_3−d_)-NiO and BCZY27(BaCe_0.2_Zr_0.7_Y_0.1_O_3−d_)-NiO were obtained by ball milling the stoichiometric amounts of oxide/carbonate precursors (BaCO_3_ 14341 by Alfa Aesar (Haverhill, MA, USA)), ZrO_2_ (40140 by Alfa Aesar (Haverhill, MA, USA)), CeO_2_ (11328by Alfa Aesar (Haverhill, MA, USA)), Y_2_O_3_ (11180 by Alfa Aesar (Haverhill, MA, USA)) and NiO (45094 by Alfa Aesar (Haverhill, MA, USA)) in deionized water for 24 h, with zirconia media. The BCZY to NiO ratio was 35 to 65 wt.%. The mixed powders were then dried and mixed with PVA (Polyvinyl alcohol, 98–99% hydrolyzed, high molecular weight by Alfa Aesar (Haverhill, MA, USA)) as a binder and Darvan 821A (Vanderbilt Minerals, CT, USA) as deflocculant in DI water. The mixture was kept on a stirring plate for 24 h.

The dip-coating solution was prepared by mixing the precursors (BaCO_3_, ZrO_2_, CeO_2_, Y_2_O_3_) and 0.5 wt.% NiO as sintering aid with polyethylene glycol (PEG) and Heraeus V006 (Heraeus, USA) in ethanol. The solution was stirred for several days on a magnetic stirring plate.

The casting was performed by filling the mold with the slurry for a casting time of 6 min, maintaining the level of the slip to the top of the mold in order to compensate for the loss of water during the slip casting process. The mold was inverted, and the excess slurry was poured out. After the tube was removed from the mold (24–48 h of drying), the surface was gently hand-burnished and cleaned with a tack cloth. The tube was afterward dipped in the dip coating solution for 2 s and set to dry for 15 min, with three successive coatings applied.

The tubes thus prepared were sintered in air in a vertical orientation with the open end placed on a bed of BCZY-NiO powder. A burnout step was first achieved by heating the sample slowly (60 °C/h) to 450 °C with a 1 h dwell at 450 °C. The furnace was then heated to 1550 °C for 5 h with a heating/cooling rate of 200 °C/h.

A portion of the bottom of each sintered tube was sawed off for the sealing experiment. A paste of copper–silver braze was prepared by mixing Ag with 1 mol% TiO_2_ and 4 mol% CuO [36] and 3 wt.% ethylcellulose in α-terpineol. The paste was mixed in an agate mortar until it was homogenous. The tubes were either sealed onto an alumina disc or a BCZY disc prepared by solid-state reactive sintering as described in [14]. The brazing paste was painted onto the two surfaces to join and set to dry in the air. The sample was then slowly heated to 450 °C (60 °C/min) with 1 h dwell to burn out the organics and then to 1080 °C (200 °C/min) for 10 min. Some of the brazes were covered with a Resbond (940 HT) layer that was sintered for 1 h at 800 °C with a slow heating rate up to 450 °C (60 °C/h), with 1 h dwell time, and then heating and cooling rates of 200 °C/h.

The gas tightness was evaluated on a reduced BCZY-Ni/BCZY tube (5%H_2_ in Ar for 24 h at 600 °C) after it was sealed onto a BCZY washer with a hole in the middle, which was itself brazed onto a ½-inch (1.25 cm) alumina tube. A tube compression fitting using a Teflon ferrule was used on the alumina tube to feed 2 bar of He/Ar.

Co-sintering of the symmetric cell, BCZY-NiO/BCZY/BCZY-NiO, has generated considerable interest for galvanic hydrogen pumping [22], as it reduces the number of high-temperature heat treatments down to one. The outer electrode for symmetric cells was prepared by painting a BCZY-NiO layer onto the unfired electrolyte (mixture of precursors (BaCO_3_, ZrO_2_, CeO_2_, Y_2_O_3,_ and NiO) with 3 wt.% of ethylcellulose in α-terpineol). After sintering (5 h at 1550 °C), the specimen was reduced to 5% H_2_ in Ar for 24 h at 600 °C.

Cross-sections of the support tubes, coated tubes, and sealed junctions were observed using an FEI Quanta 600i Environmental Scanning Electron Microscope (ESEM) by FEI Company (Hillsboro, OR, USA) or a JEOL JSM-7000 Field Emission Scanning Electron Microscope (FESEM) by JEOL (Tokio, Japan). Both secondary electrons (SE) and back-scattered electron (BSE) images were collected. In some cases, the samples were embedded in epoxy, burnished with fine-grit sandpaper, and then with diamond paste until a 1-micron finish was obtained. The polished samples were then cleaned in ethanol in an ultrasonic bath for 5 min.

## 3. Results

### 3.1. Slip Casting and Co-Sintering

Figure 2a shows a picture of BCZY27-NiO tubes as slip casts, i.e., before sintering. Six-centimeter-long tubes with a wall thickness of one millimeter could easily be obtained. Dip-coated tubes are illustrated in Figure 2b,c, before and after sintering, respectively. Additional coating layers led to the peeling of the electrolyte layer. It is clear from Figure 2c that the bottom of the tube flared during the sintering process. This part was cut after sintering with a diamond saw. The shrinkage rate of the co-sintered tubes was between 22 and 23%.

Figure 3 shows the micrographs of a BCZY27-NiO tube coated with three layers of BCZY27 and co-sintered at 1550 °C, with two different magnifications and a BCZY18-NiO tube coated with three layers of BCZY18 at the highest magnification. It may be observed that the applied BCZY thin film is homogeneous and approximately 20 microns in thickness. As solid-state reactive sintering was used for the co-sintering of the support and the thin membranes, the interface is smooth.

A top view of the BCZY27-NiO/BCZY27 tube is reported in Figure 4 to confirm the well-sintered microstructure of the electrolyte.

### 3.2. Brazing

The first brazing experiments were performed by joining a BCZY18-NiO/BCZY18 co-sintered tube onto a piece of dense alumina. The micrographs from Figure 5 represent the secondary electron images (on the left) and the back-scattered electron images (on the right) at two different magnifications. The light and dark phases in the support are BCZY18 and NiO, respectively. The BCZY18 film was well-adhered to the BCZY18-NiO support. The Ag braze wetted and surrounded the BCZY18 film and joined the tube onto the alumina setter. A strong bonding was obtained.

These encouraging results motivated us to test the hermeticity of the brazes. For that purpose, a BCZY27-NiO/BCZY27 tube was sintered and then brazed onto a BCZY washer with a 5 mm hole in the center. The washer was then sealed onto an alumina tube that was used for feeding gases. The sealed tube was reduced in order to generate open porosity in the Ni-BCZY support. As observed from Figure 6, the tubes could be successfully joined to the washer. This brazing required several heatings, as two different brazes were required. This led to Ag dripping down the sides. Two bar of He/Ar were fed into the inside of the tubes. Tiny bubbles were observed at the brazing. A layer of Resbond (940 HT) was then applied on top of the braze, and the hermicity was checked again. No visible bubbling was observed when maintaining 2 bar in the inside of tubes.

### 3.3. Symmetric Cells

Symmetric cells (BCZY-NiO/BCZY/BCZY-NiO) were prepared by painting a BCZY-NiO outer layer onto the dip-coated electrolyte. The challenge with the painting of this electrode was the potential for damaging the unsintered electrolyte layer, which could allow the outer electrode to contact the inner electrode of the support. Figure 7 is an example of a symmetric cell before and after sintering. After reduction, the electrolyte was shorted, preventing any further electrical testing of the symmetric cells.

## 4. Discussion

### 4.1. Challenges and Advantages of the Slip Casting and Solid-State Reactive Sintering of Short Tubular Half-Cells

As mentioned in the results section, the number of dip coatings was limited to three, as additional coatings led to the peeling of the electrolyte layer as the coating was drying. Figure 8a shows an example of the electrolyte layer peeling off after the fourth dip coating. Such a peeling appears due to surface tension. Surface modification was attempted by painting a BCZY-NiO layer on the slip cast tube (mixture of precursors (BaCO_3_, ZrO_2_, CeO_2_, Y_2_O_3_, and NiO) with 3 wt.% of ethylcellulose in α-terpineol), but no significant improvement was observed. An example of a slip cast tube with a hand-painted BCZY-NiO layer is displayed in Figure 8b.

The closed-end tubes described in this paper were stand-fired, with the open end facing down. Above an aspect ratio of about 10 for length to diameter, the tubes must be hang-fired.

From the list of processes to fabricate tubes (Table 1), the other technique to make closed-end tubes is the coating of the BCZY/NiO material onto a glass test tube [30]. The current research highlights the advantages of using slip casting (an easy, fast, and cost-effective technique) combined with solid-state reactive sintering (a cost- and time-effective method since only one high-temperature heat treatment is needed). The fabrication of the one-piece mold (Figure 1) avoids the presence of a seam on the cast tubes due to a mold parting line. It, therefore, limits the polishing of the cast tube prior to the electrolyte deposition. Other advantages of fabricating these molds are: (1) the size flexibility: the size is dictated by the arbor used during the mold fabrication; (2) the shaping versatility: this study focused on tubes, but more complex-shaped pieces were also slip cast, such as an eight-lobe tube. Some know-how was necessary to unmold the parts, and further research will focus on a better mold design.

It is clear that fabricating closed-end tubes lowers the number of required seals, as only one side needs to be sealed. When extruding the tubes, one end of the tube is usually caped [22,23,25,26,28]. The other end of the tube can be brazed to an alumina tube, as exemplified in this study, or could be sealed to the testing device with a graphite ferrule and matching Swagelok tube fitting.

### 4.2. Reactor Design Based on Short Tubular Cells

Protonic-ceramic membrane reactors can potentially improve any chemical or catalytic process that benefits from hydrogen separation and operates at temperatures around 600 °C. For example, the effectiveness of steam electrolysis, hydrocarbon reforming, chemical synthesis, and electrochemical compression can all be improved with protonic ceramics [7,22,23,24,25,26]. The cost-effective, short, closed-end tubes offer new opportunities for reactor configurations, compared with those using long extruded tubes or planar bipolar stacks. Of course, details of reactor layouts, catalysts, and operating conditions depend on the targeted process.

As discussed in the foregoing text and illustrated in Figure 9, the short tubes are based on a relatively thick (≈1 mm) porous composite Ni-BCZY support that serves electrochemically as the negatrode (fuel electrode). The Ni phase serves as the electron-conducting phase, the BCZY serves as the proton-conducting phase, and gases fill the pore volumes. The relatively thin (≈20 µm) BCZY dense electrolyte membrane is dominantly a proton conductor, but it is a mixed ionic–electronic conductor (MIEC) that also conducts oxygen vacancies and small polarons [37]. The composition and structure of the outer relatively thin (≈20–50 µm) porous electrode (positrode) depends on the desired chemical or electrochemical process. In steam electrolysis, for example, the positrode could be a mixture of BCZY and LSCF ((La,Sr)(Co,Fe)O_3−d_). It could also be a so-called triple-conducting oxide (TCO) such as BCFZY (Ba(Co,Fe,Zr,Y)O_3−d_) [17]. Figure 9 shows a wire-wrap external current collector. However, the figure is silent on the internal (negatrode) current-collection strategy. It may be noted that tubular internal current collection can be challenging, but there are numerous innovative strategies being developed. A major advantage of using short tubes is the short negatrode current collection path.

Figure 9 offers the general outlines of a possible concept. In this case, a bank of tubes is positioned onto a manifold header, with process gas flowing around the tube exteriors. Consider, for example, steam electrolysis with the steam (process gas) being introduced from the top. Steam reacts electrochemically on the tube exterior electrode (positrode) to produce protons that enter the protonic ceramic membrane and deliver electrons to the external circuit. The protons are driven across the membrane (Nernst-Planck flux), where they react on the interior electrode (negatrode) with electrons from an external circuit to produce H_2_. The hydrogen exits each tube through the supporting manifold. The gas-phase O_2_ that is produced from the electrolysis mixes with steam in the space surrounding the tube bank, with the resulting mixture being exhausted from the reaction chamber. The mixture composition outside the tubes depends on the feed flow rates and the H_2_ production rates. By controlling the exhaust H_2_ back pressure, some electrochemical compression is possible [38].

## 5. Conclusions

Slip cast tubes were prepared with the precursors to form BaCe_x_Zr_0.9−x_Y_0.1_O_3−d_ (BCZY)-NiO tubes, with x = 0.1 and 0.2, referred to as BCZY18 and BCZY27, respectively. An electrolyte layer of the corresponding BCZY composition was deposited on the unsintered tube by dip-coating (three layers). Upon solid-state reactive sintering, a thin (20 microns) dense uniform BCZY membrane was well-adhered to the substrate. These tubes were successfully brazed onto alumina and BCZY by brazing (silver, with TiO_2_ and CuO additions), followed by a coating with a Resbond layer. This study highlights the two main advantages of slip casting: its size and shape versatility.

Symmetric cells (BCZY-NiO/BCZY/BCZY-NiO) were prepared from co-sintering the three layers. Unfortunately, the electrolyte was damaged, leading to shorts upon the NiO reduction and therefore resulting in untestable cells.

New opportunities for reactor configurations are feasible with these cost-effective, short, closed-end tubes. A versatile reactor design is proposed, with possible applications in steam electrolysis, hydrogen separation, or catalytic membrane rectors, depending on the choice of the outer electrode (positrode).

## Figures and Tables

**Figure 1 membranes-12-00242-f001:**
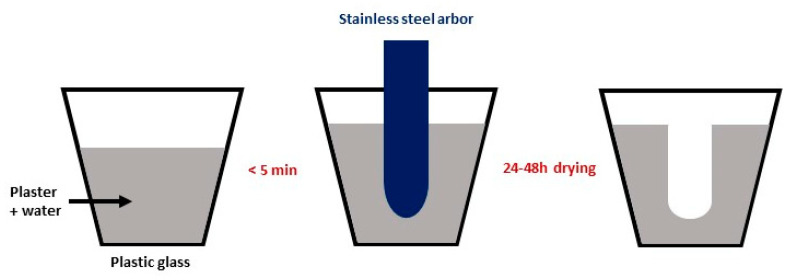
Fabrication of the molds for slip casting.

**Figure 2 membranes-12-00242-f002:**
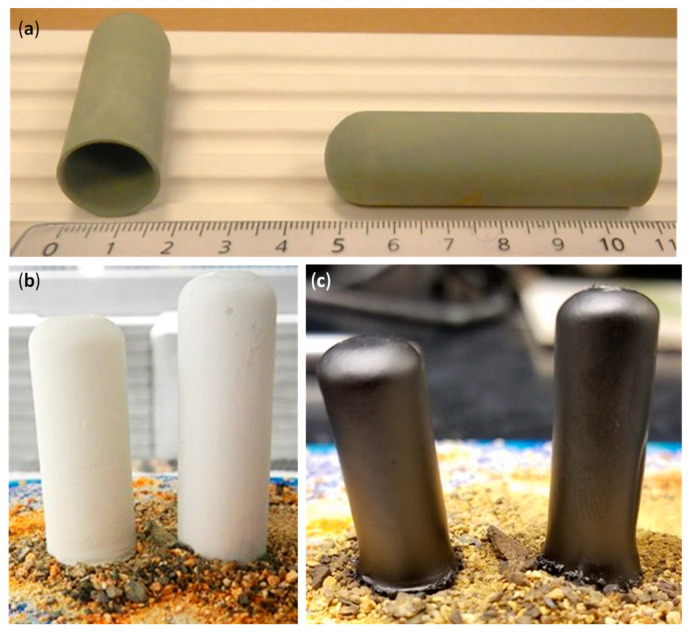
BCZY27-NiO slip cast tubes: (**a**) after casting, (**b**) after dip-coated the BCZY27 electrolyte, and (**c**) after co-sintering.

**Figure 3 membranes-12-00242-f003:**
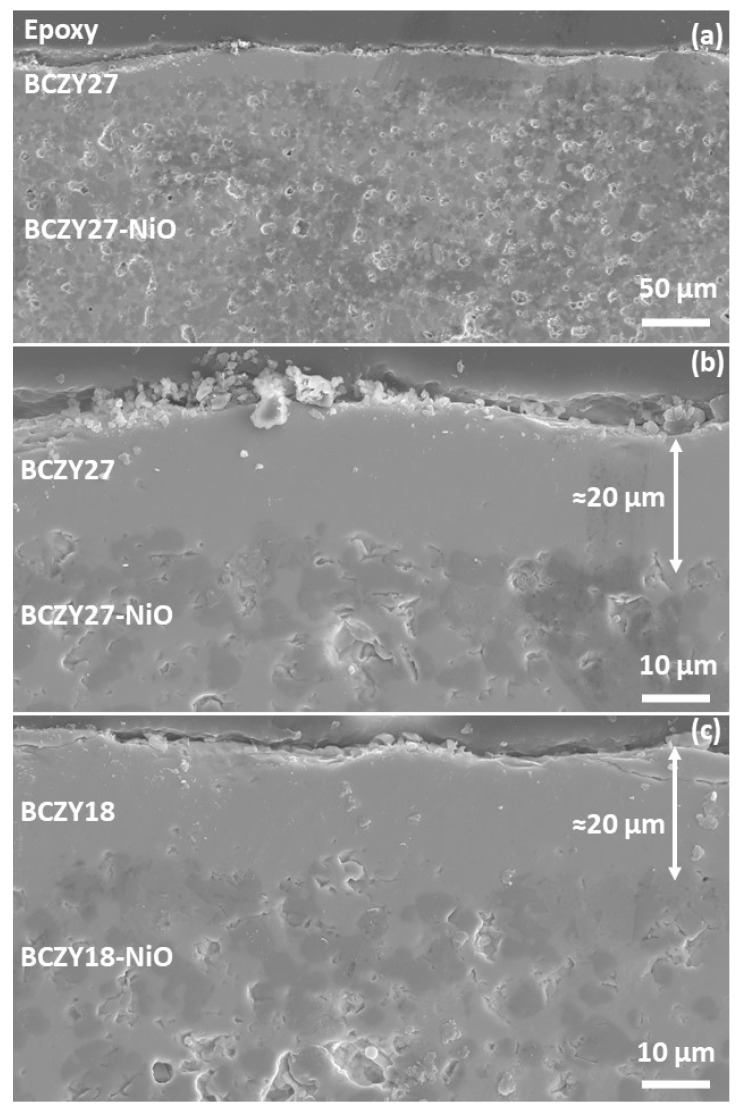
Secondary electron micrographs of a polished cross-section of (**a**,**b**) a co-sintered BCZY27-NiO/BCZY27 tube and (**c**) a co-sintered BCZY18-NiO/BCZY18 tube.

**Figure 4 membranes-12-00242-f004:**
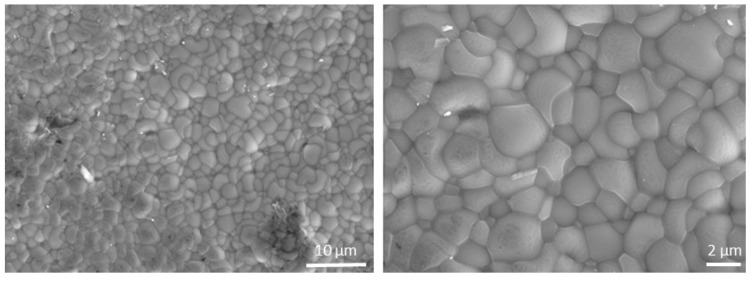
Secondary electron micrographs of the surface of the BCZY27 electrolyte from a co-sintered BCZY27-NiO/BCZY27 tube.

**Figure 5 membranes-12-00242-f005:**
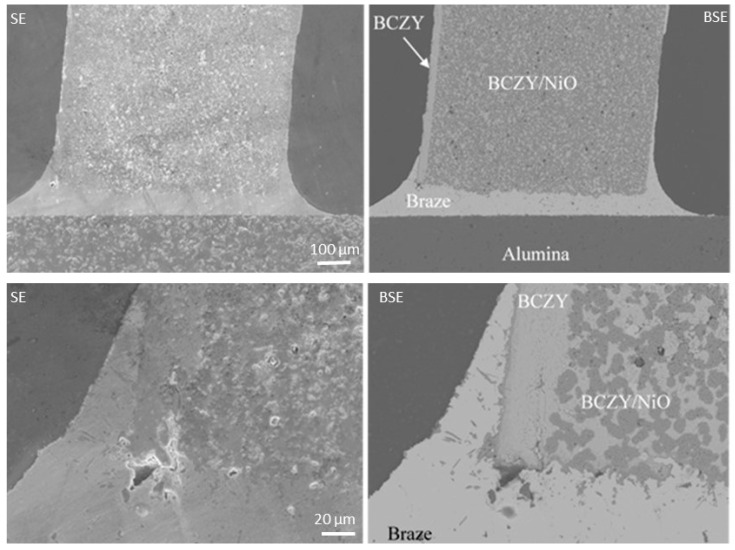
Secondary electron images (SE, on the left) and the back-scattered electron images (BSE, on the right) of a BCZY18-NiO/BCZY18 brazed onto an alumina piece at two different magnifications.

**Figure 6 membranes-12-00242-f006:**
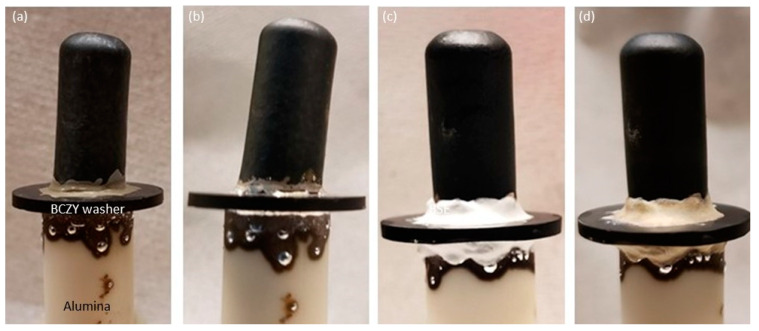
BCZY27-NiO/BCZY27 sintered tube on a BCZY holed washer and alumina tube: (**a**) after painting the braze paste, (**b**) after sintering the braze paste at 1080 °C, (**c**) after painting the Resbond layer onto the brazes, and (**d**) after sintering the Resbond layers at 800 °C.

**Figure 7 membranes-12-00242-f007:**
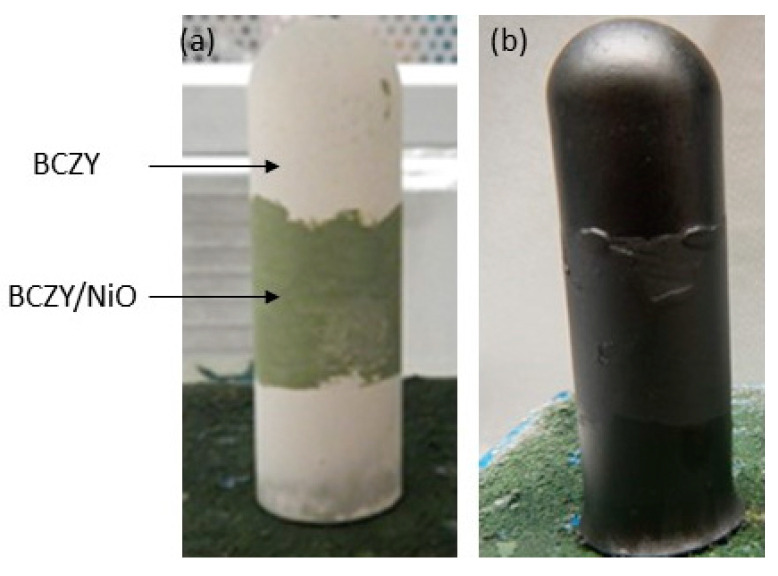
BCZY27-NiO/BCZY27/BCZY27-NiO symmetric cell tube: (**a**) before and (**b**) after sintering.

**Figure 8 membranes-12-00242-f008:**
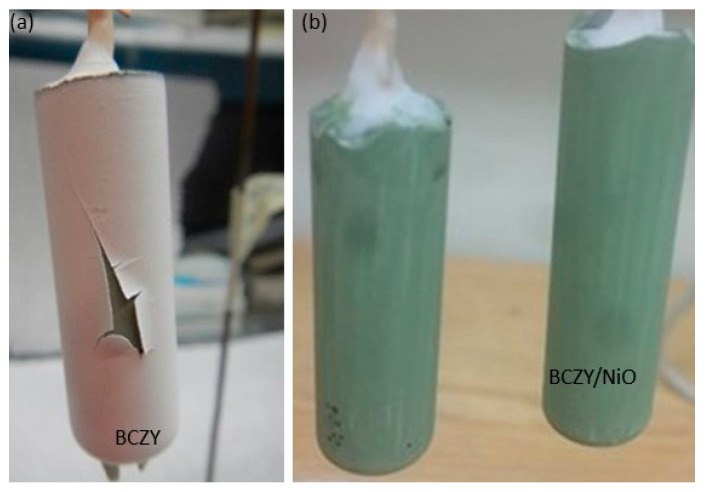
(**a**) BCZY27-NiO tube after the fourth dip coating of the electrolyte layer, (**b**) BCZY27-NiO intermediate layer (ethylcellulose in α-terpineol) painted onto the slip cast tube, prior to slip casting.

**Figure 9 membranes-12-00242-f009:**
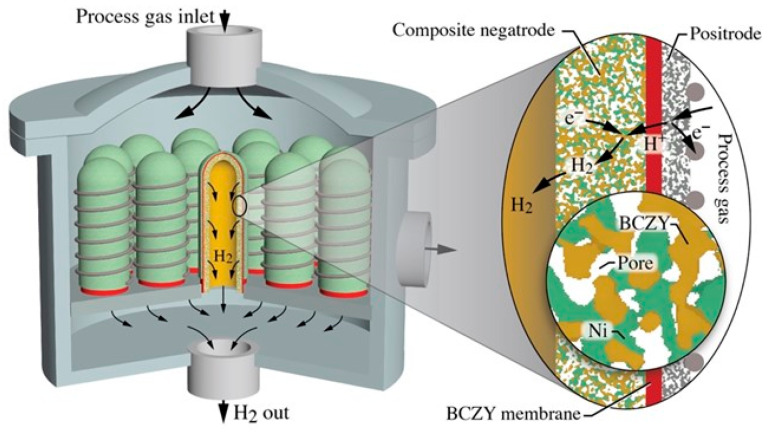
Cartoon illustration of a possible protonic ceramic membrane reactor concept. A bank of short, closed-ended tubes is bonded to a collection manifold. A steady flow of process-gas mixture enters at the top. Flowing process gases and reaction products occupy the reactor volume outside protonic ceramic tubes and exhaust laterally. The expanded figure on the right illustrates details of the tubes’ membrane-electrode assembly and its electrochemical function.

**Table 1 membranes-12-00242-t001:** Relevant literature data on tubular protonic-ceramic-based devices.

Composition	Fabrication	Reference
BCZY27-NiO/BCZY27	Extrusion BCZY27-NiO Spraying BCZY27 co-SSRS Caping of the tube	[23,25,26]
BCZY27-NiO/BCZY17	Extrusion BCZY27-NiO Spraying BCZY17 co-SSRS Caping of the tube	[22]
BCZY27-NiO/BCZY18	Extrusion BCZY27-NiO Spraying BCZY1 8co-SSRS Caping of the tube	[22]
BCZYYb7111-NiO/BCZYYb7111	Phase inversion BCZYYb7111-NiO Dip coating BCZYYb7111 Conventional co-sintering Open-end tube	[27]
BCZDy-NiO/BCZDy	Tape-calendering BCZDy-NiO and BCZDy Conventional co-sintering Caping of the tube	[28]
BCZYYb7111-NiO/BCZYYb7111	Extrusion BCZYYb7111-NiO Brushing BCZYYb7111 co-SSRS Open-end tube	[29]
BCZY72-NiO/BCZY72	Coated on a glass test tube BCZY72-NiO and pre-sintering Dip-coating BCZY72 Conventional co-sintering Closed-end tube	[30]
BCZY71-NiO/BCZYZn	Extrusion BCZY71/NiO pre-fired Dip coating BCZYZn Conventional co-sintering Open-end tube	[31]
BCZY27-NiO/BCZY27	BCZY27/NiO laser 3D printing BCZY27 spray coating co-SSRS Open- or close-end tube	[32]

BCZY27: BaCe_0.2_Zr_0.7_Y_0.1_O_3−d_, BCZY18: BaCe_0.1_Zr_0.8_Y_0.1_O_3−d_, BCZY17: BaCe_0.1_Zr_0.7_Y_0.2_O_3−d,_ BCZYYb7111: BaCe_0.7_Zr_0.1_Y_0.1_Yb_0.1_O_3−d,_ BCZDy: BaCe_0.5_Zr_0.3_Dy_0.2_O_3−d_, BCZYZn: BaCe_0.7_Zr_0.1_Y_0.16_Zn_0.04_O_3−δ_, SSRS: Solid-state reactive sintering.

## Data Availability

The data presented in this study are available on request from the corresponding author.

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
