# Peer review of "Slip Casting and Solid-State Reactive Sintering of BCZY(BaCexZr0.9−xY0.1O3−d)-NiO/BCZY Half-Cells"

_membranes, 2022, doi:10.3390/membranes12020242_

Round 1
Reviewer 1 Report
Please find my comments below:
- The abstract should be briefly written to describe the purpose of the research, the principal result, and major conclusion. Authors should revise it.
- In the Introduction, explain which is the different between extrusion and slip casting processes with supported by literatures review.
- Symbol of temperature degree, without underline.
- Standardization of units of measurement.
- In line 60, from where you got the PVA material.
- Authors should add the SEM of BCZY18.
- If possible, adding the thickness of layers on SEM Figures.
- In Title of Figures 3, 4 and 5, you mention to use Secondary electron micrographs and in text you mention used SEM, please confirm which one are you used.
Author Response
The authors acknowledge the reviewers for their time and corrections. The manuscript has been modified accordingly. The responses to the comments are given below in blue font, and the revisions are highlighted in yellow in the manuscript.
Reviewer 1:
1. The abstract should be briefly written to describe the purpose of the research, the principal result, and major conclusion. Authors should revise it.
The abstract has been revised with more details about the presented results.
2. In the Introduction, explain which is the different between extrusion and slip casting processes with supported by literatures review.
A paragraph about the extrusion and slip-casting processes is added in the introduction, together with three references.
One of the most common fabrication methods for the tubular support electrode is extrusion, as it leads to a uniform density distribution [33]. The BCZY powder (or the mixture of precursors for solid-state reactive sintering) and NiO are mixed with additives (dispersants, binders, plasticizers) and a solvent. The as-prepared paste is then extruded through a die (typically cylindrical) and dried [34]. The resulting tube is open on both ends. To prepare closed-end tubes, slip casting is a cost-effective technique. The first step consists in making a mold in gypsum. The water-based ceramic slip is then poured into the mold, and the water is sucked out by capillarity. After drying, the ceramic body can be removed from the mold and sintered [35].
3. Symbol of temperature degree, without underline.
The symbol for the temperature has been corrected.
4. Standardization of units of measurement.
The units of measurement have been corrected to metric.
5. In line 60, from where you got the PVA material.
The provider for the PVA is provided.
6. Authors should add the SEM of BCZY18.
A micrograph of the BCZY18 electrolyte onto the BCZY18-NiO support is added as Figure 3c.
7. If possible, adding the thickness of layers on SEM Figures.
The thicknesses of the electrolyte membrane have been added in figure 3.
8. In Title of Figures 3, 4 and 5, you mention to use Secondary electron micrographs and in text you mention used SEM, please confirm which one are you used.
All the micrographs in Figures 3 and 4 were taken with secondary electrons, while the micrographs from figure 5 are either from secondary electrons or back-scattered electrons. It is now clarified in the experimental section.
Both secondary electrons (SE) and back-scattered electrons (BSE) images were collected.
Reviewer 2 Report
The manuscript is an interesting piece of work, worthy of being published in Membranes. English is acceptable. Results are clearly and concisely discussed. Conclusions are supported by the presented data. References are adequate. Hence. I recommend the manuscript publication in Membranes.
Author Response
The authors acknowledge the reviewers for their time and corrections. The manuscript has been modified accordingly. The responses to the comments are given below in blue font, and the revisions are highlighted in yellow in the manuscript.
The manuscript is an interesting piece of work, worthy of being published in Membranes. English is acceptable. Results are clearly and concisely discussed. Conclusions are supported by the presented data. References are adequate. Hence. I recommend the manuscript publication in Membranes.
We thank the reviewer for the positive feedback. The manuscript was double-checked for typos and misspellings.